# Green Synthesis of Advanced Carbon Materials Used as Precursors for Adsorbents Applied in Wastewater Treatment

**DOI:** 10.3390/ma16031036

**Published:** 2023-01-24

**Authors:** Georgeta Predeanu, Valerica Slăvescu, Marius Florin Drăgoescu, Niculina Mihaela Bălănescu, Alexandru Fiti, Aurelia Meghea, Petrisor Samoila, Valeria Harabagiu, Maria Ignat, Ana-Maria Manea-Saghin, Bogdan Stefan Vasile, Nicoleta Badea

**Affiliations:** 1Research Center for Environmental Protection and Ecofriendly Technologies, University POLITEHNICA of Bucharest, 1-7 Gheorghe Polizu Str., 011061 Bucharest, Romania; 2SC Cosfel Actual SRL, Griviței Rd., 95-97, Sector 1, 010705 Bucharest, Romania; 3“Petru Poni” Institute of Macromolecular Chemistry, 41A Grigore Ghica Voda Alley, 700487 Iasi, Romania; 4National Research Center for Micro and Nanomaterials, University POLITEHNICA of Bucharest, 6 Iuliu Maniu Bdv., 061344 Bucharest, Romania

**Keywords:** vegetable waste, microwave heating, advanced carbon materials precursors, adsorption, pollutants removal

## Abstract

Huge amounts of vegetable waste, mainly resulting from the food industry, need large areas for storage, as they could cause hazardous environmental impact, leading to soil and water pollution or to CO_2_ emissions during accidental incineration. This work was aimed at recycling certain lignocellulosic waste (walnut shells, kernels of peach, apricot, and olive) to design advanced carbon material precursors (ACMP) to be used for obtaining nano-powders with high applicative potential in pollution abatement. Both waste and ACMP were characterized using proximate and elemental analysis, and by optical microscopy. Complex characterization of raw materials by FTIR, TGA-DTG, and SEM analysis were carried out. The ACMP were synthetized at 600–700 °C by innovative microwave heating technology which offers the advantages of lower energy consumption using 3.3 kW equipment at laboratory level. The ACMP ash < 3% and increased carbon content of 87% enabled the development of an extended pore network depending on degassing conditions during heating. TEM analysis revealed a well-developed porous structure of the synthesized ACMP carbonaceous materials. Due to the presence of oxygen functional groups, ACMPs exhibit adsorption properties highlighted by an iodine index of max. 500 mg/g and surface area BET of 300 m^2^/g, which make them attractive for removal of environmental pollutants such as dyes having molecule sizes below 2 nm and ions with pore dimensions below 1 nm, widely used industrially and found in underground waters (NO_3_^−^) or waste waters (SO_4_^2−^).

## 1. Introduction

As a result of rapid development of urban areas, industry, and the increase in the transportation sector, environmental pollution encompasses nowadays large areas and represents one of the main concerns of the contemporary world, with all the environmental factors—air, water, and soil—being widely affected [1]. Specific measures to decrease greenhouse gas (GHG) emissions are needed. Among the main recommendations is the preservation of the forestry fund.

The inefficient management of wastewaters and of solid waste increase the organic and inorganic pollutants released in surface and ground waters. In this way, a major water deficit concerning both quantity and its quality are predicted, so that by 2025 circa 49% of the world population will have a limited access to water sources.

All these concerns are clearly highlighted within the United Nations Environmental Program launched by the United Nations Organization (UNEP) in 1988 referring to environmental protection as planetary ecosystem and universal heritage. The main priorities include mitigation of the GHG emissions, assuring water requirement and its quality as a primordial resource, waste recycling, and reuse as renewable resources for various applications using clean technologies [2]. Indeed, many adequate technologies have been developed as efficient methods for carbon capture, pollutant removal, and waste storage, but their use is rather limited by cost and energy requirements. Alternatively, adsorption has been proven to be a promising environmentally friendly technique due to its low cost, low energy requirement, and large applicative potential over a wide range of temperature and pressure.

Carbon based adsorbents are among the main materials involved in the adsorption process offering application efficiency through their characteristics [3,4,5]. For liquid-phase applications, advanced carbon materials (ACMs) are by far the best-known adsorbents of this type, whose upward trend is rapidly strengthening due to the need to implement environmental regulations [6,7,8,9,10,11,12,13]. In this regard, some carbon-based nanomaterials have recently been obtained due to their diversity, favorable properties, and active applications including fullerenes, graphene-based materials (playing an important role in energy conversion and storage fields due to the attractive qualities of graphene) [14,15], carbon nanotubes (considered as highly conductive materials for battery electrodes), carbon nanofibers, which are attracting great attention in the electroanalytical area for energy storage, etc. [16,17,18,19,20].

Another application direction is represented by modified activated carbon (AC) materials, such as AC supported Pd-Cu catalysts embedded on fibrous cloths applied for continuous water denitrification [21], or other noble metal catalysts on AC (Pt/AC and Ru/), heavy metal catalysts (Cu/AC, CoMo/AC, Mo/AC, Mn/AC, Fe/AC) [22,23], or sludge-based AC produced during a recycling process [24].

As there is a growing demand for ACMs on the international market, the classical raw materials used, such as coals, walnut or coconut shells, sawdust, and hardwood, are continuously decreasing.

Worldwide, the available amounts of lignocellulosic materials for AC production are estimated at the following levels: hardwood (*Fagus*, *Carpinus betulus*)—20,000,000 t/year, coconut—450,000 t/year, sawdust—150,000 t/year. As 500 m^2^ of the forest is cut to obtain one tone of AC, alternative resources of raw materials, including various agricultural waste have been identified, such as: pineapple waste (for adsorbents for methylene blue removal from wastewater) [25,26]; coconut shell and rice husk [27], and palm shells [28]. At the same time, the mechanism of the adsorption process using AC has been developed to quantify adsorption efficiency [29], including competitive adsorption in multicomponent systems [30].

The authors have recognized expertise in AC manufacturing from fossil wood (xylite) and fruit kernels by carbonization through conventional electric heating technology and their utilization in wastewater treatment, demonstrating large surface area and high porosity in such biomass-derived carbon systems [31,32,33,34,35,36,37,38,39,40].

Fruit kernels represent a viable alternative raw material for obtaining ACs and there is a significant potential for fruit kernel waste at the European level, including cost benefits. Referring to peach kernels, the production of the leading five countries (Italy, Spain, Greece, France, and Turkey) is evaluated at circa 310,000 t/year, out of which if only 30% were to be reused, would result 20,000–30,000 t/year AC. For apricots and plums kernels, the production of the leading five countries (Serbia, Germany, Poland, Spain, and France) is estimated at 100,000 t/year, out of which, with 30% reuse would result 10,000 t/year AC. For Romania, the total fruit kernels of plums, peaches, and apricots reused in a similar proportion would provide about 1000–2000 t/year AC.

The requirement to obtain sustainable products during the last decades has encouraged the development of sustainable technologies characterized by a reduction in energy consumption [41].

One of the technologies to produce ACs is carbonization in the microwave field, based on the thermochemical decomposition of organic material in an inert atmosphere [42]. Microwaves are electromagnetic waves in the wavelength range between 1 mm and 1 m, with corresponding frequencies between 300 GHz and 300 MHz. Industrial microwave heating applications use frequencies of 915 MHz and 2450 MHz respectively [43].

The heating mechanism consists of the penetration of microwaves into the raw material particles, the microwave energy being transformed into thermal energy by dipole oscillation (liquid medium) or by interfacial polarization (solid medium). Compared to conventional heating, in microwave heating the temperature gradient is transferred from inside to the outside of the particle. These different heating and reaction mechanisms provide advantages for microwave-assisted pyrolysis (MAP) over conventional electric pyrolysis, i.e., lower thermal inertia, faster response speed, higher heating rate, higher heating efficiency, directed heating, etc. [44].

The increased interest in MAP is highlighted by the typology, respectively the multitude of published topics, including the following: fundamentals, principles, and mechanisms of MAP [45]; MAP techniques used for waste conversion or treatment [45]; physical and chemical properties of bio-oils obtained by conventional pyrolysis and MAP of biomass materials [46]; yield and properties of biochar obtained from conventional pyrolysis and MAP of plant biomass [46]; effects of feedstock characteristics and pyrolysis parameters on bio-oil production from MAP of plant biomass [47]; reactor configurations and kinetic models for MAP of plant biomass [48]; effect of operating conditions such as MW power, pyrolysis temperature, pyrolysis time, particle size, type of purge gas and MW absorbers (catalysts) on biofuel production [49,50]; reactor configurations and kinetic models for MAP of biomass [22]; socio-economic and environmental implications of biomass MAP [51].

MAP relies significantly on the interactions between microwave irradiation and the raw material used. Product yields (e.g., bio-oil, biogas, biochar) obtained from MAP also strongly depend on the electrical properties of the feedstocks [52]. In general, biomass waste has a lower absorption of microwaves, thus delaying the heating processes. In these conditions, microwave amplifiers (strong susceptors) are used, involving fast pyrolysis assisted by microwaves [53]. The physical–chemical properties of biochar can be related both directly and indirectly to the way it influences the different systems it encounters, in the various industrial applications such as: water treatment, soil remediation, catalytic systems, pharmaceuticals, medicinal products, etc. [46,54,55].

In this context, the aim of the current paper was to recycle four fruit wastes such as walnut shells, and kernels of peach, apricot, and olive. These raw materials were characterized using physical–chemical and complex instrumental techniques, while a green innovative microwave heating technology was developed, as it offers the advantage of lower energy consumption. In addition, one of the purposes of this work was to design and obtain an ACMP to be further used for producing several powders with high applicative potential in pollution reduction. ACMP exhibits promising adsorption performances as indicated by material textures, pore structure distribution, and specific surface area.

## 2. Materials and Methods

### 2.1. Sample Origin and Characterization

Four types of lignocellulosic waste of scientific interest (kernels, shells) originating mainly from fruit processing such as peach, apricot, and olives, as well as walnut shells, were characterized, Table 1.

An amount of 100 g of each waste type was selected and prepared for the laboratory analysis in accordance with SR ISO 1988:1996 [56], SR ISO 5069-2:1994 [57].

The wastes were characterized by determining the total moisture (STAS 5264:95) [58] and the moisture analysis (SR ISO 331:1994) [59] that preceded the other proximate characteristics was reported at anhydrous state (dry basis).

Ash was determined according to SR ISO 1171:1994 [60], volatile matter and fixed carbon according to STAS 5268: 1990 [61]. Volumetric density was determined on the raw material as received, dried in air, and slightly pressed, while the real density followed STAS 816: 1981 [62]. For the preparation of pellets for the microscopic study, samples having <1 mm grain size were embedded in epoxy resin and polished with different grain sizes of carborundum paper and alumina according to ISO 7404-2:2009 [63]. The determination of the elemental analysis was carried out with a C, H, N, O, S analyzer with accessories type FLASH 2000.

### 2.2. Fourier Transform Infrared (FTIR) Spectroscopy Analysis

To investigate the surface chemistry the waste samples were analyzed by Fourier transform infrared spectroscopy (FTIR) technique. Spectra were recorded on a Bruker Vertex FTIR spectrometer (Manning Road Billerica, MA, USA), resolution 2 cm^−1^, in the range 4000–400 cm^−1^ (Sigma-Aldrich Chemie GmbH, Taufkirchen, Germany) by the KBr pelleting technique.

### 2.3. Thermal Analysis ATG

Thermal analysis ATG was used to study the mass losses occurring through the thermal decomposition of waste in the temperature range 30–700 °C, with a heating rate of 10 °C/min in nitrogen atmosphere (flow of 50 mL/min). The thermal degradation was followed on a STA 409 PC Luxx Simultaneous thermal analyzer.

### 2.4. Scanning Electron Microscopy (SEM) Analysis

By means of the SEM technique (scanning electron microscopy) the morphological characteristics of the four lignocellulosic wastes were investigated using a Verios G4 UC scanning electron microscope (Thermo Fisher Scientific, Bremen, Germany). This type of investigation aimed at identifying and highlighting the important morphological aspects specific to each waste, and to use this information to study the morphology influence on the pore size of ACMP.

The preparation of the raw material for SEM analysis proceeded through several steps, as follows: 1. Washing—samples were washed 3 times with a sufficient volume of distilled water (1 L water/100 g waste, under agitation (for 15 min) at a temperature of 40 °C; 2. Drying—samples were dried in a heating oven at 105 °C for 24 h after which they were transferred to a desiccator; 3. Primary crushing; 4. Grinding—using the Pulverisette 11 mill in 3 stages, (i) 5 min at 3500 rpm for 12 s and a 20 s break, (ii) 5 min at 5500 rpm for 15 s and a 60 s break, (iii) 5 min at 9000 rpm for 20 s and a 60 s break, 5. Sizing—for 15 min using the Retsch AB 200 system (amplitude 55) on the following sieves: 4 mm; 2 mm; 1 mm; 0.5 mm; 0.25 mm; 0.125 mm. The fraction between 125 µm and 250 µm was used for the analyses.

### 2.5. Transmission Electron Microscopy (TEM) Analysis

To investigate the morphology of the prepared lignocellulosic waste-derived carbonaceous materials at the nanoscale, transmission electron microscopy was considered. Thus, a Hitachi High-TechHT7700 transmission electron microscope (TEM, Hitachi High Technologies Corporation, Tokyo, Japan) was used for registering micrographs for each sample. The equipment was operated in high-contrast mode at 120 kV accelerating voltage. Before analysis, all samples were ground and dispersed in ethanol by ultrasonication, and subsequently a few drops were placed on formvar-coated copper grids. The copper grids thus prepared were then dried overnight at 333 K.

### 2.6. BET Analysis and Adsorption Processes

To quantitatively investigate the porous structure of the obtained ACMP, the N_2_ adsorption technique was used to identify the pore structure distribution and specific surface area. The adsorption processes were performed through adsorption/desorption isotherms measurements based on adsorption/desorption of N_2_. The specific surface area of the material was determined by nitrogen adsorption–desorption measurements, using a NOVA 2200e Quantachrome instrument (Quantacrome Instruments, Boynton Beach, FL, USA), working at a nitrogen liquefaction temperature of −196 °C. Before measurements, all samples were outgassed at room temperature for two hours, to evacuate physically adsorbed molecules and to empty the pores. The BET (Brunauer–Emmet-Teller) equation was applied to the isotherm (in the region of P/P0 = 0.05–0.35) to evaluate the BET specific surface area. Further, the BJH (Barrett–Joyner–Halenda) theory was used to estimate the pore size distribution and to determine the mean pore diameter. The total pore volume was calculated taking the nitrogen adsorbed volume at relative pressure 0.95 P/P0.

### 2.7. Pyrogenation of Waste Biomass by Microwave Heating

The ACMP samples were obtained by pyrogenation of waste biomass in a microwave heating laboratory equipment which was designed and built with a frequency of 2.45 GHz and having a microwave power of 3.3 kW. The microwave heating process was carried out in an inert atmosphere and was monitored by a temperature controller. The hot gases were collected through a system consisting of an expansion vessel, a condenser, and a liquid phase collector vessel, Figure 1.

### 2.8. ACMP Synthesis

For the synthesis of ACMP and considering the distribution of the electric field in the resonant cavity of the installation, the waste amount used varied between 550–700 g. For the correct measurement of the temperature (max. 1100 °C) using a K-type thermocouple and for a better contact with the sample, the walnut shells and kernels were hand crushed. After crushing, adequate size fractions in the range 2–10 mm were selected. A total of 10 samples of four lignocellulosic wastes were studied for physical–chemical, porosimetric, and complex characterization, Table 2.

## 3. Results and Discussion

### 3.1. Proximate and Ultimate Analysis Results

Proximate analysis reveals the qualitative uniformity of the raw materials (Table 1). Thus, their high purity—a characteristic which is compulsory for obtaining high-grade adsorbents—is revealed by ash that varies in the range 0.38–0.86% having a mean value of 0.63%. Volatile matters, which also include those of the kernel’s core, are quite similar (mean value 78.75%) varying in the range 77–81% and of the fixed carbon ranges in the range 18–22% with an average of 20.6%. The volumetric density, parameter which is important to ensure an adequate loading of the heating reactor during the synthesis experiments, showed also quite similar values (mean 0.397 g/cm^3^) for all the wastes that were hand crushed.

Regarding the elemental analysis, it was observed that the waste had a high carbon (C) content with a maximum value for the case of apricot kernels (51.12%) and of 47.15% for nutshell (Table 1). The oxygen (O) content was present in a significant proportion, at a maximum in walnut shells (41.97%) and followed closely by peach and olive kernels (39–40%), then apricot kernels (38%). In the case of nitrogen (N) content, walnut shells had the highest values of 2.1%. Hydrogen (H) content showed a slight variation being maximum for the apricot kernels (6.95%) and minimum for the peach kernels (5.94%). Sulphur (S) was absent from all wastes tested.

### 3.2. Optical Microscopy

The petrographic analysis outlines the relationships between the microtexture (morphology) of the particles and waste type, optical appearance, and size of their micropores. The identification of the morphological aspects of the native material that will be subjected to thermal treatments provides in the micrographs in Figure 2 important information regarding the appearance of the biomass waste. A well-defined morphology of the raw materials can be seen, in which the cellulose microfibers of the biomass can be identified.

More specifically, microscopic analysis reveals the following: (i) the composition of the cell walls, which consists of cellulose, lignin, hemicellulose, and secondary components (resins, fats, tannin, dyes, etc.); (ii) the walls of the woody cell are made up of agglomerations of cellulose macromolecules organized in micelles and lamellae of submicron dimensions; (iii) the rows of mycelia form voids, in which lignin is usually stored, with a cementing role.

These morphology characteristics will be later completed with the details regarding the quality of ACMP obtained by microwave processing.

### 3.3. Complex Characterization of the Wastes

#### 3.3.1. Fourier-Transform Infrared (FTIR) Spectroscopy Analysis

There are major compositional differences between walnut shells and peach, apricot, and olive kernels, in the sense that they all have a woody part, theoretically composed of mixtures of cellulose, hemicelluloses, and lignin, and the kernel samples have, in addition to the shell, a complex mixture of organic compounds grouped into two large classes—polyphenols and terpenes. The specific FTIR spectra of the four samples are presented in Figure 3.

From the relative intensities of the different bands, several compositional differences can be established as follows: walnut shell is characterized by a lower content of ester groups (bands at 1742 and 1744 cm^−1^) [64,65,66,67,68]; the amount of oxidized aliphatic groups (such as acids, ketones, aldehydes) is higher in walnut shells (considering the bands at 1618 and 1659 cm^−1^) [67,69]; there are fewer aromatic groups in walnut shells (bands at 1508 and 1510 cm^−1^) [67,68,69], as compared to kernel samples. On the other hand, there are similarities between the spectra of the analyzed kernels samples. Thus, the bands around 3005 cm^−1^ suggest the presence of unsaturated fatty acids and/or epoxy groups [70,71,72]. The kernel samples present ester and carboxylic groups considering the bands at 1746 and 1659 cm^−1^, respectively [68]. Likewise, the olive kernel sample indicates a higher content of aromatic compounds than the apricot kernel sample, comparing the band at 1515 cm^−1^ and the bands at 1539 and 1518 cm^−1^, respectively [66,73,74].

#### 3.3.2. TGA-DTG Results and Thermal Decomposition Behavior

Figure 4 shows the thermal decomposition curves (TG) and (DTG) of the target waste, and Table 3 shows the main thermal parameters.

The pyrolysis of walnut shells (Figure 4a) takes place in three main stages of decomposition, with the specification that the second stage is divided into two or even three phases of decomposition because of the overlap of several thermal processes. For the first degradation stage (30–170 °C) the relatively low mass losses (3.01%) are determined by the removal of moisture and some volatile extracts that are organic compounds of small molecules (oils, resins, etc.). In the second stage (170–430 °C) significant mass losses occur when the decomposition of the hemicellulose and cellulose as major components of the biomass begins. First, hemicellulose decomposition takes place (T_onset_ = 207 °C) with mass losses of approx. 5%.

The process continues much more intensively, with the increase of temperature, due to the beginning of the thermal degradation of the cellulosic component (considering the shoulder from 254 °C). The mass losses at this stage are significant (around 52%), also noted by the maximum rate of thermal decomposition of 5.70%/min (T_peak_ = 339 °C). This aspect indicates that, in addition to the elimination of the side functional groups of the two polymers, the breaking of intramolecular or intermolecular hydrogen bonds, generating water, and even the breaking of glucoside chains and their fragmentation, also occur. According to previous findings, the molecular structure of the cellulose is altered, and the degree of polymerization decreases from 10,000 to about 200 causing the production of carbon residue [75].

The further increase in temperature results in the thermal decomposition of lignin, which is a complex aromatic component based on phenolic compounds (p-coumaryl alcohol, coniferyl alcohol, and synaptic alcohol) and which is much more thermally stable. The presence of a shoulder at 254 °C is noted which can be attributed to the initiation of the lignin decomposition process which continues up to 700 °C. It is important to note that the mass losses related to this process are around 12%.

In the last two stages, in addition to the mass losses shown, there are several decomposition reactions of the material retained in the carbonic residue, formed by aromatization reactions, breaking glycosidic bonds, carbon–hydrogen bonds, carbon–oxygen bonds, methoxyl groups, as well as the fragmentation of the units obtained by the depolymerization of the three biomass polymers [76]. The thermal stability of the walnut shells was analyzed according to mass losses of 10% (T_10_), respectively 20% (T_20_) and it was found that the temperatures at which these losses were recorded were 253 °C, respectively 287 °C.

Thermal behavior of kernel samples is like that observed for walnut shell samples (Figure 4b–d) [75,76,77,78]. Thus, thermal decomposition is also characterized by three main areas corresponding to the following processes: (i) dehydration, (ii) active pyrolysis of hemicelluloses and cellulose with the release of volatiles and carbon formation; (iii) passive pyrolysis where lignin degradation takes place accompanied by the combustion of hydrocarbons, reactions crosslinking/aromatization, and char formation.

As can be seen from Table 3, the values of the decomposition temperatures and the percentage of volatiles in each stage are dependent on the sample nature. Thus, it was found that the kernel samples contain smaller amounts of water than the samples of walnut shells. The start of stage II occurs at the lowest temperature for apricot kernels (163 °C), while the other two kernel samples start to decompose at temperatures higher than 230 °C; for olive and peach kernels 231 °C and 242 °C, respectively.

From the volatile percentage values for stages II–IV, the following series can be compiled depending on the content of the three main components of the samples:-hemicellulose content: walnut shells < apricot kernels ≈ peach kernels ≈ olive kernels-cellulose content: apricot kernels < olive kernels < peach kernels << walnut shells-lignin content: walnut shells < peach kernels < olive kernels < apricot kernels

It is important to note that the carbon residue values are in the range of 23–29%, the highest being that of the peach kernel sample.

#### 3.3.3. Scanning Electron Microscopy (SEM) Analysis

In the case of the powder obtained from walnut shells, the agglomeration tendency is shown in Figure 5a. A well-developed porous structure of carbonaceous materials from walnut shells can be achieved by directing temperatures in the range of 500–800 °C, according to the literature [79]. The sample of peach kernels (Figure 5b) is morphologically characterized by an agglomeration with a high degree of compaction. Certain smaller spherical structures can also be observed. The sample does not show a visible native porous structure, and this fact can be explained by the pores clogging after the grinding process.

In the literature, Šoštarić et al. [70] mention that the reason why the pores of raw peach kernel material can become clogged is due to the compression of the material during milling which occurs during this process [79,80]. For apricot kernels (Figure 5c), the surface morphology is characterized by the existence of smaller spherical aggregate agglomerations. The photomicrographs show the presence of typical porous morphology of biomass waste having pores of approximately 1 μm. The relevant factors for obtaining an adequate porosity within the activated carbons mentioned in the literature are the temperature, heating rate, and pyrolysis cycles through which the decomposition of the organic compounds of the raw material takes place [81]. The olive kernel samples are morphologically characterized by compact agglomerations (Figure 5d). The literature mentions that the use of olive kernels in the synthesis of activated carbon allows materials with pores of different sizes and shapes to be obtained depending on the temperature and the duration of the heat treatment [74].

### 3.4. Synthesis of ACMP by Microwave Process

During the synthesis of advanced carbon material precursors by pyrogenation, at temperatures below 400 °C, changes in the chemical structure of lignocellulosic waste and cumulative weight losses are mainly due to the elimination of water, and carbon monoxide/dioxide. These changes have the effect of free carbon atom aromatization and condensation processed that group in regularly arranged crystalline structures.

These transformations can be assimilated to wood carbonization, considering their vegetable origin. Thus, during the carbonization of wood—the first stage of the process—primary char and tars are generated. This is followed by the second stage, the reactions in which the primary tar breaks down into secondary tar, char, and gases. In addition to decomposition reactions, other chemical reactions take place such as condensation, polymerization, reduction, etc. At 400–500 °C the edges of the C–C crosslinked structures start to lose their stability due to polymerization. One part remains unsaturated, and the others react with H_2_, becoming saturated. The first volatilize as tar vapors and the others condense in the form of charcoal with a high content of aromatic hydrocarbons [35,82,83].

The pyrolysis tests for obtaining ACMP were carried out in different operating conditions, in which the following parameters were changed: the heating rate (which varied from ˂15 °C/min, 15–30 °C/min, 30–40 °C/min, up to max. 113 °C/min), the microwave irradiation time during soaking, and the operating temperature (600–700 °C). All tests were performed at atmospheric pressure.

A number of ten pyrolysis tests (T1–T10) were run. The microwave-assisted operation parameters for the pyrolysis process of the used lignocellulosic waste are presented in Table 4 and the thermal regime of the heating stage in the microwave-assisted pyrolysis process is presented in Figure 6.

The yield and characteristics of the pyrolysis products are strongly influenced by the operating conditions (e.g., temperature, heating rate) and the properties of the feedstock. The average yields obtained from ACMP synthesis of 26% biochar, (Figure 7 and Figure 8A) are comparable to those mentioned in the literature of 27% in the case of wood [84].

The lignocellulosic nature of the waste favors obtaining by conventional pyrolysis global yields (biochar 30%, tar 42% and gases 28%) such as those obtained in the case of wood (biochar 34%, tar 42%, and gases 24%), (Figure 8); the use of this feedstock being considered appropriate to obtain ACMP by pyrogenation technology.

### 3.5. ACMP Characteristics

#### 3.5.1. Physico–Chemical

The variation in ash is quite low and is reflected by the small value of the standard deviation, the range varying from a minimum of 1.2% (sample 8), maximum of 3% (sample 4), and an average of 2.2% values. This reveals the feedstock highly suitable for further steps of activation.

The average content of volatile matter of 8.52% reveals the adequate degree of degassing of the material during ACMP synthesis in the microwave field. The recorded values show a significant variation between the minimum value 4.48% for sample 6, and respectively the maximum value of 17.51% for sample 1.

The values recorded by the fixed carbon vary between 80.32% (sample 1) and 93.43% (sample 6). The average value of 89.28% reveals a substantial enrichment in carbon through pyrogenation, higher than 79% [84] or 74.3% mentioned for agricultural waste by certain publications [85]. Figure 9 shows a strong correlation between the fixed carbon and volatile content (correlation index R^2^ = 0.99), Table 2. The values recorded by the real density parameter (g/cm^3^) vary between 1.4287 g/cm^3^ (sample 1) and 1.614 g/cm^3^ (sample 6) with an average value of 1.5412 g/cm^3^. The values recorded by the volumetric weight parameter (g/cm^3^) vary between 0.31 g/cm^3^ (sample 5) and 0.389 g/cm^3^ (sample 3) with an average value of 0.3558 g/cm^3^. There are also changes of nitrogen, carbon, hydrogen, and oxygen contents against feedstock.

#### 3.5.2. Transmission Electron Microscopy (TEM) Analysis

Further, to find out more structural information, transmission electron micrographs were undertaken. Thus, Figure 10 shows TEM images revealing the morphology of the prepared carbonaceous materials from walnut shells, peach, apricot, and olive kernels. As can be seen, all TEM images indicate the development of a porous structure of mainly amorphous carbon phase, confirmed by SAED patterns (insets in Figure 10), with regions displaying structure ordering, most probably due to the formation of graphene layers [86]. Moreover, the SAED patterns also reveal the presence of graphite crystals or another form of carbon with a lower degree of order.

#### 3.5.3. Adsorption Characteristics

ACMP adsorption characteristics determined by chemical (iodine index) and physical (N_2_ adsorption) analyses are presented.

The average value of the iodine index is 406.85 mg/g, the minimum value being 274.89 mg/g (sample 1), respectively, a maximum of 488.37 mg/g (sample 8). The iodine index shows the same or increased values as other biochar produced in the range 500–700 °C from various feedstocks mentioned in the literature, some made from commercial cellulose with iodine adsorption capacities of 371.40 mg/g [87]. This revealed the following aspects:-Physical-textural structure and increased purity of waste (ash < 1%), determines the adsorption capacity.-Porosity depends both on the native structure of the raw material, on the temperature and microwave heating rate, and synthesis duration.-Low heating rates lead to an increased number of fine pores, which, favors the product quality.

The textural properties investigation of ACMP samples was conducted by determining the specific surface area, porosity, pore and micropore volume. This revealed the following aspects: most of samples show levels that correlate satisfactorily with the S_BET_, with values of 114 m^2^/g as minimum and of 295 m^2^/g as maximum. A relatively lower surface area varying between 20–80 m^2^/g was reported for vegetal wastes by some authors [88]. The literature mentions large cylindrical pores with diameters of 5–40 μm and S_BET_ of 181 m^2^/g that were developed in the case of wood biochar due to the vascular cell structure of the parent biomass, over a temperature range around 700 °C with a heating rate of 10 °C/min [84]. Other authors reported an S_BET_ of 134 m^2^/g for biochar from fruit waste made at 600 °C [89].

In Figure 11 the Langmuir isotherms (A1, B1) are presented and the corresponding pore size distribution (A2, B2) of ACMP made from walnut shells (A) and peach kernels (B) obtained when the adsorption is restricted to the monolayer level. According to IUPAC classification [90], the registered isotherms are of type I (b) which usually is found in carbonaceous materials having pore size distributions in the micro–meso range (wider micropores and narrow mesopores). Thus, the adsorbent seems to have micropores that are too small to support the adsorption of nitrogen molecules in more than one layer, as well as the capillary condensation in the pores. Both ACMP samples (walnut shells and peach kernels) exhibit the same comparable type I (b) isotherm.

The pore size distributions in the micro–meso range (wider micropores and narrow mesopores) developed in ACMP by microwave heating allow the possibility of assessing these materials for the intended applications. The results are promising regarding the ACMP testing for purification of waters infested for example with certain dyes having molecule sizes in the range of the ACMP pore dimensions, below 2 nm, such as: methylene blue, rhodamine, methyl orange, or others. In addition, several tests are foreseen regarding removal of ions with pore dimensions below 1 nm such as nitrate and sulphate ions widely used industrially and found in underground waters (NO_3_^−^) or waste waters (SO_4_^2−^).

In Table 5 a preliminary statistical analysis of the data sets is shown.

The correlation coefficients (Table 5) indicate quite good correlation values among several parameters, such as: S_BET_ and iodine index (0.845), S_µ_ (0.767), V_µ_ (0.752), and V_tot_ (0.808). The equation obtained is as follows (Figure 12):S_BET_ = 1.2598 Iodine index − 355.41(1)

The model shows a very high adjusted R—square coefficient (R^2^ = 0.7816).

The dependency between the parameter S_BET_ and S_µ_ is described by a linear model with a high correlation coefficient value of 0.94 while for the relationship between V_µ_ and V_tot_, a direct strong connection is concluded (correlation coefficients are very high 0.93, respectively 0.96). A strong correlation is also revealed among three parameters S_µ_ and V_µ_ of 0.99 and V_tot_ of 0.90 while for the linear models presented above, a multiple linear regression model was tested.

#### 3.5.4. Microscopic Characteristics

The petrographic study of ACMP samples was aimed at the qualitative determination of structural composition and intragranular porosity in the case of samples obtained on a 3.3 kW laboratory equipment. The research on the microstructure highlighted several aspects of theoretical and practical interest regarding the microwave synthesis process, such as evolution and efficiency, as well as the characteristics of the pyrogenated product, the ACMP.

Synthetically, and with specific reference to the most representative analyzed samples, the most frequent microstructural images of the ACMP granules presented can be correlated with the adsorption capacity towards iodine and the S_BET_ developed during degassing at 600–700 °C.

Thus, from the photomicrographs presented in Figure 12, the following general aspects result:ACMP has a specific microstructure, characterized by a generally small and very small porosity, located in the walls of the larger pores.ACMP microstructures are specific to the different constituents of the original biomass.The cellular structure mainly shows rounded pores.The very fibrous structure, finely porous, with rounded pore walls, reveals links through radial chains with elongated pores.The structural composition of the granules that form the walls of the degassed pores is noticeable, and in some samples, there is a very high total porosity.

The photomicrographs presented in Figure 13 provide information on the following: (i) how the microwave heating process is carried out and the phenomena that occur during pyrogenation; (ii) the specificity of the structures and the correlation with the morphological type of the granules from lignocellulosic biomass; (iii) the type and size of the porosity, which influences the susceptibility of ACMP as an effective adsorbent in this stage of processing the target waste, through the value of the SBET of 295 m^2^/g.

## 4. Conclusions

The high purity of the investigated waste (ash below 1%), allows the development of promising ACMP characteristics which are necessary in the further procurement of high-grade adsorbents. The petrographic analysis outlines the relationships between waste type and particle morphology. Optical aspects of the native material provide information regarding the appearance of the biomass waste, in which lamellae of submicron dimensions and agglomerations of cellulose macromolecules, organized in micelles, can be identified that form voids where lignin with a cementing role is usually stored.

FTIR spectroscopy and thermogravimetric analysis data demonstrated the existence of complex mixtures of hemicelluloses and lignocelluloses for most components of the analyzed samples. From the analysis of the TG/DTG curves, it was found that the investigated samples present both similarities and differences. The thermal curves of the four samples were characterized by three main zones assigned to: dehydration, active pyrolysis of hemicelluloses and cellulose with the release of volatiles and the formation of carbon, and passive pyrolysis where the degradation of lignin takes place accompanied by the combustion of hydrocarbons, crosslinking/aromatization reactions and char formation. SEM investigation highlighted morphology characterized mainly by the existence of spherical aggregates or compact agglomerations.

The pyrolysis tests for obtaining ACMP were carried out in a microwave heating laboratory equipment with a frequency of 2.45 GHz having a microwave power of 3.3 kW and a maximum temperature of 600–700 °C undern different operating conditions. The parameters were changed as follows: the heating rate, microwave irradiation time during soaking, and the operating temperature. As a result of microwave heating technology, global yields, such as those obtained in the case of conventional heating applied to vegetable waste and wood, were obtained.

TEM analysis revealed a well-developed porous structure of the synthesized ACMP carbonaceous materials. TEM images accompanied by SAED patterns proved the porous structure of the mainly amorphous carbon phase, which presents from place-to-place graphite crystals or another form of carbon with a lower degree of order.

The investigation of textural properties of ACMP powder fine samples was carried out by determining physical–chemical and optical investigation, such as iodine adsorption, specific surface area, porosity, and pore volume, and identifying the micropores, including use of optical microscopy. The ACMP ash in the range 2–3% and increased carbon content of 87% allow the development of an extended pore network depending on degassing conditions during heating. Due to the presence of oxygen functional groups, ACMP exhibited adsorption properties highlighted by an iodine index of max. 500 mg/g and surface area BET by 300 m^2^/g. The dependency between the parameter S_BET_ and S_µ_ was described by a linear model with a high correlation coefficient value of 0.94, while for the relationship between V_µ_ and V_tot_, a direct strong connection was concluded (correlation coefficients are very high 0.93, respectively 0.96).

The registered isotherms are of type I (b) which usually is found in carbonaceous materials having pore size distributions in the micro–meso range (wider micropores and narrow mesopores). Thus, the adsorbent seems to have micropores that are too small to support the adsorption of nitrogen molecules in more than one layer, as well as capillary condensation in the pores. The pore size distributions in the micro–meso range (wider micropores and narrow mesopores) developed in ACMP by microwave heating give the possibility of testing these materials for the intended applications.

The morphological aspects of ACMP identified by optical microscopy can be correlated with the iodine adsorption capacity developed during degassing at 600–700 °C. ACMP has a specific microstructure, characterized by a small porosity created in the wider pore walls, with a cellular structure consisting of rounded pores. The increase of intragranular porosity occurs in parallel with the reduction of the carbon matrix and of the volume of the pore walls. The photomicrographs highlight both the characteristics of porous texture and the types of carbon matrix, which can contribute, in certain proportions, to the dimensions of the adsorption surface, and in the subsequent phases of ACMP activation.

The results are promising regarding ACMP utilization for purification purposes. Some dyes such as: methylene blue, rhodamine, and methyl orange having molecule sizes below 2 nm can be removed by using ACMPs with certain pore sizes. Further tests are thought necessary regarding removal of ions with pore dimensions below 1 nm, such as nitrate and sulphate ions widely used industrially and found in underground waters (NO_3_^−^) or waste waters (SO_4_^2−^).

## Figures and Tables

**Figure 1 materials-16-01036-f001:**
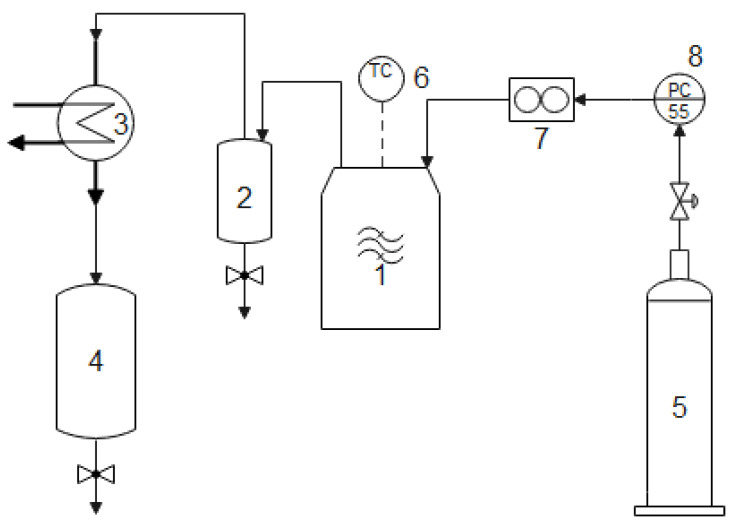
Concept scheme of the microwave heated laboratory equipment of 3.3 kW/2450 MHz: 1—microwave reactor, 3.3 kW, 2.45 GHz; 2—hot gas expansion vessel; 3—condenser; 4—liquid phase collector vessel; 5—inert gas tank; 6—temperature controller; 7—gas flow meter; 8—pressure controller.

**Figure 2 materials-16-01036-f002:**
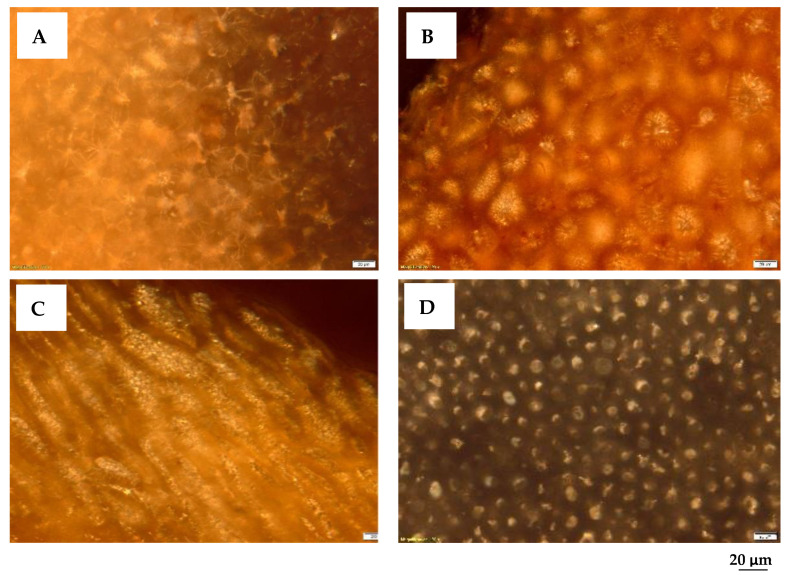
Photomicrographs of biomass wastes. Reflected light, immersion, 200×: (**A**). Walnut shells. (**B**). Peach kernels. (**C**). Apricot kernels. (**D**). Olive kernels.

**Figure 3 materials-16-01036-f003:**
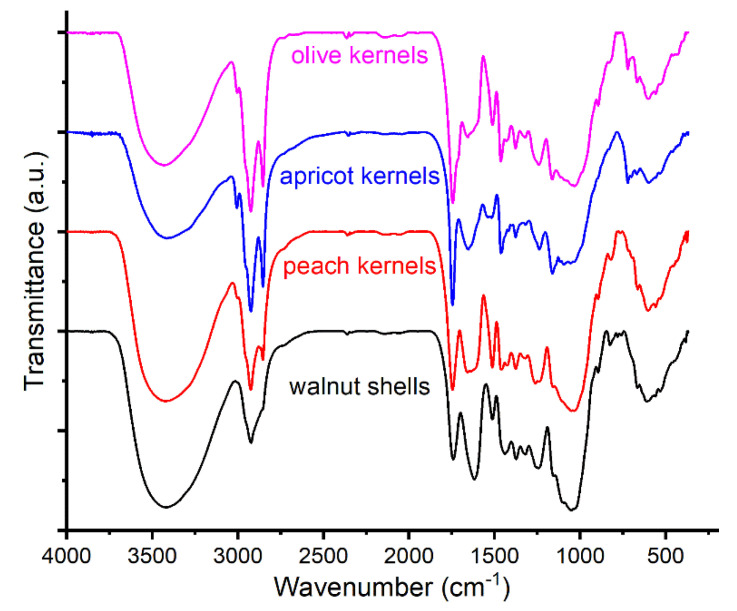
FTIR spectra of walnut shells and kernels of peach, apricot, and olive.

**Figure 4 materials-16-01036-f004:**
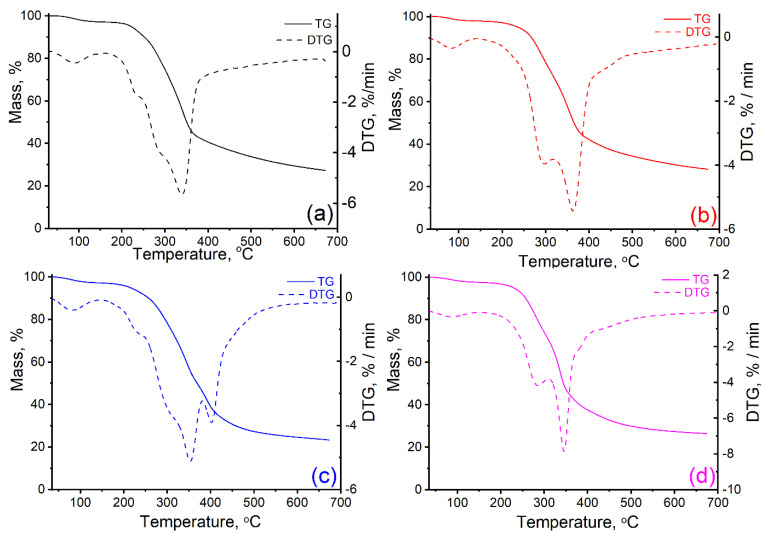
TG and DTG curves of walnut shells (**a**), and kernels of peach (**b**), apricot (**c**), and olive (**d**).

**Figure 5 materials-16-01036-f005:**
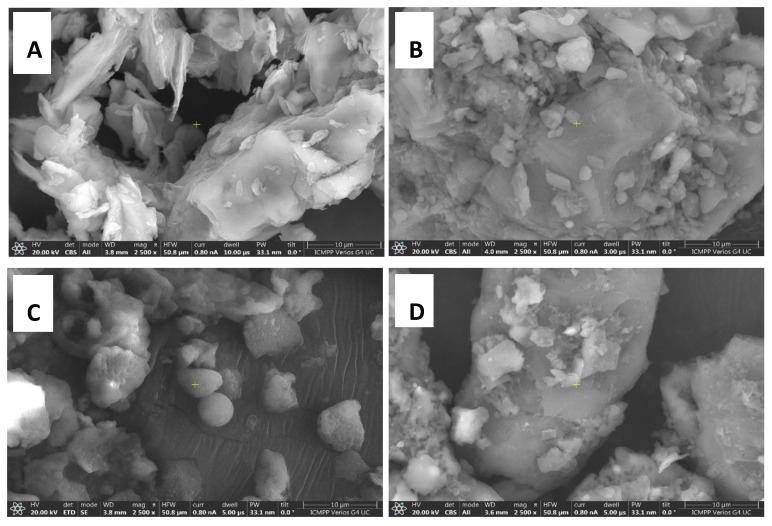
Representative SEM images of the waste type analyzed, 2500×: (**A**) walnut shells; (**B**) peach kernels; (**C**) apricot kernels; (**D**) olive kernels.

**Figure 6 materials-16-01036-f006:**
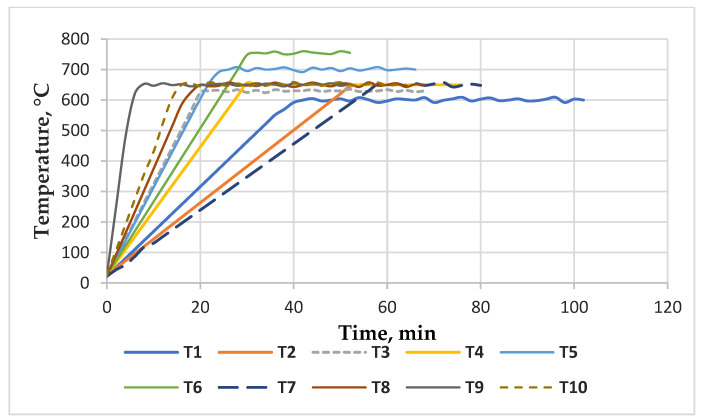
The thermal profile of the ACMP synthesis in the microwave field.

**Figure 7 materials-16-01036-f007:**
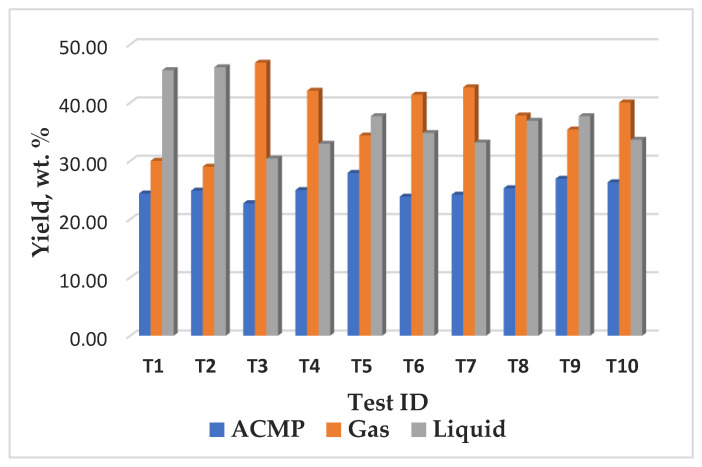
The yield of the reaction products obtained during the synthesis of ACMP in the microwave field.

**Figure 8 materials-16-01036-f008:**

Comparison between the average yields obtained in case of vegetable waste pyrogenation by microwave heating (**A**), waste by conventional heating (**B**), and wood by conventional heating (**C**).

**Figure 9 materials-16-01036-f009:**
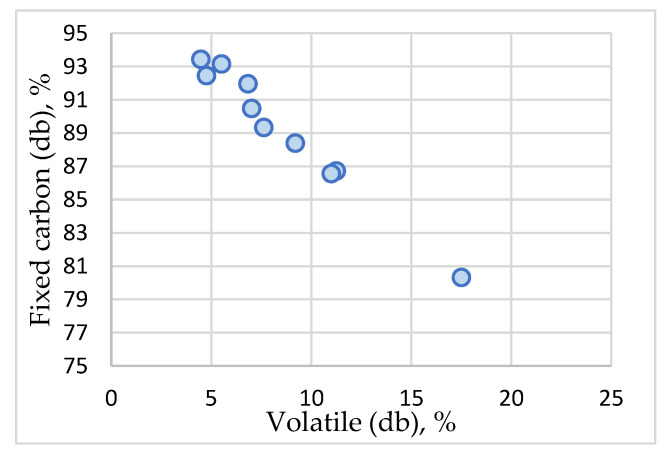
ACMP fixed carbon and volatile matter content.

**Figure 10 materials-16-01036-f010:**
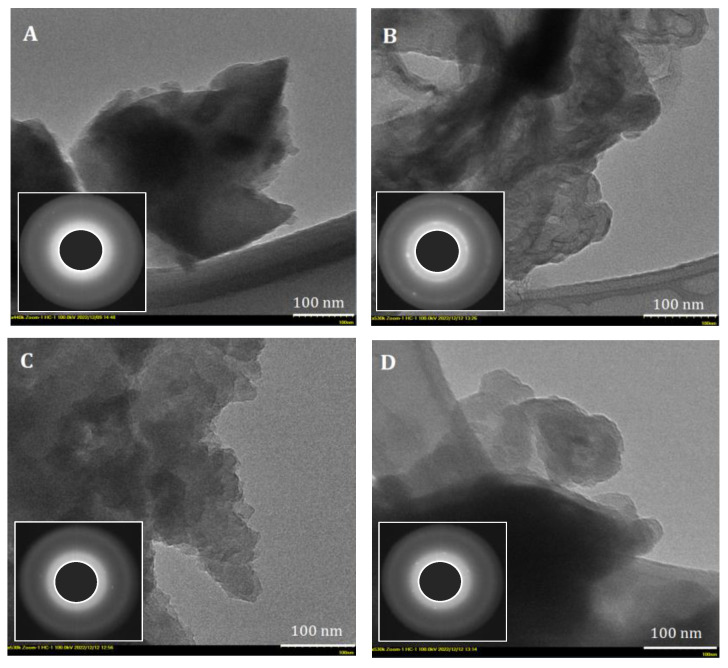
Representative TEM (bar scale 100 nm) images (inset: SAED patterns) of the ACMP type analyzed: (**A**) walnut shells; (**B**) peach kernels; (**C**) apricot kernels; (**D**) olive kernels.

**Figure 11 materials-16-01036-f011:**
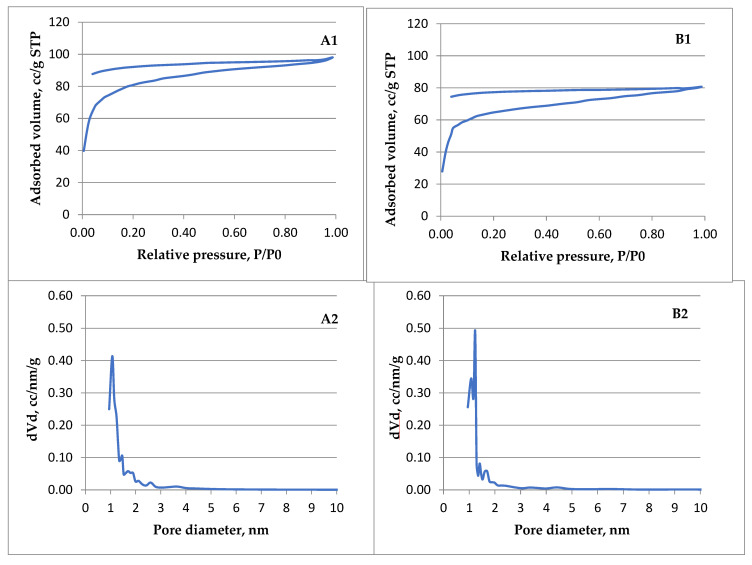
Nitrogen adsorption-desorption isotherms (**A1**,**B1**) and the corresponding pore size distribution (**A2**,**B2**) obtained for ACMP walnut samples (**A**) and peach kernels (**B**).

**Figure 12 materials-16-01036-f012:**
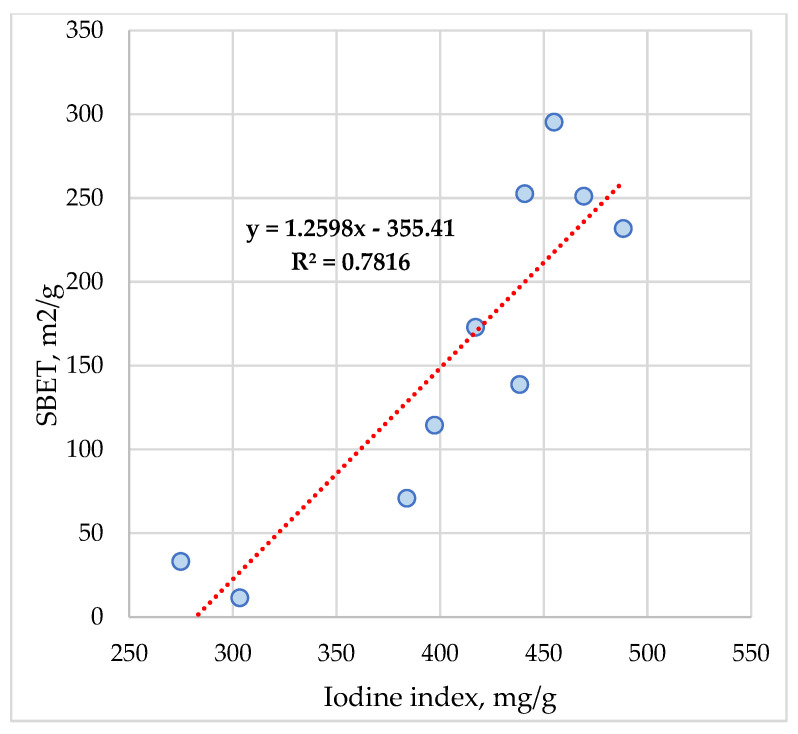
Variation between S_BET_ and iodine index of ACMP.

**Figure 13 materials-16-01036-f013:**
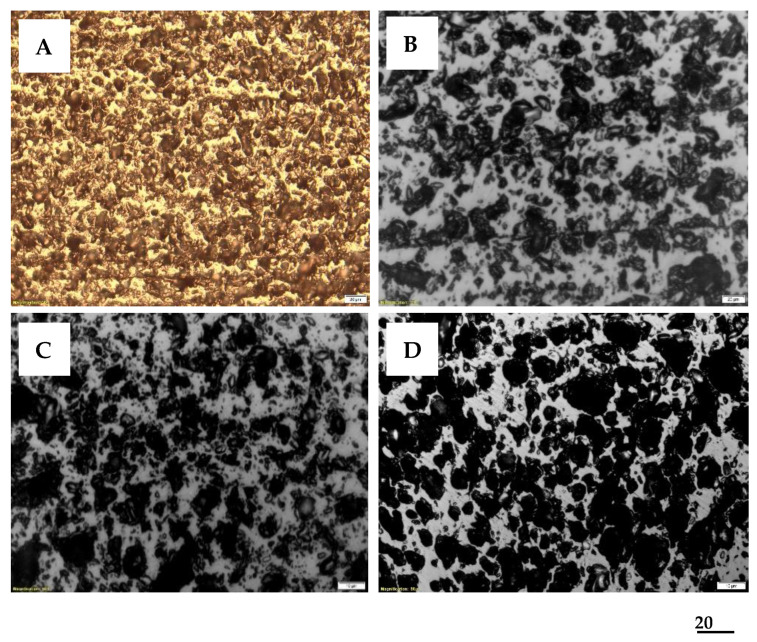
Textural and structural optical aspects typical for ACMP obtained from shells and kernels, correlated with S_BET_ and iodine number, reflected light, glycerin immersion, 200X: (**A**,**B**) fine and very fine porosity predominantly below 10 µm in ACMP of walnut shells in sample A (iodine index 455 mg/g, S_BET_ 295 m^2^/g) and sample B (iodine index 469 mg/g and S_BET_ 251 m^2^/g); (**C**) microporosity of various sizes and thicker pore walls in ACMP of peach kernels (iodine index 488 mg/g and S_BET_ 231 m^2^/g); (**D**) small porosity formed in ACMP of olive kernels (iodine index 397 mg/g and S_BET_ 114 m^2^/g).

**Table 1 materials-16-01036-t001:** Physical–chemical characteristics of biomass wastes.

Sample ID	W1	W2	W3	W4
Biomass Type/Characteristics	Walnut Shell	Peach Kernels	Apricot Kernels	Olive Kernels
Total moisture, wt%	8.61	8.21	7.59	7.92
Moisture analysis, Wa, %	2.35	3.44	1.97	1.19
Ash (db), %	0.71	0.38	0.86	0.60
Volatile (db), %	78.25	78.70	80.77	77.29
Fixed carbon (db), %	21.04	20.92	18.37	22.11
C, %	47.149	47.906	51.124	47.995
H, %	6.039	5.938	6.948	6.369
N, %	2.163	1.140	1.605	0.585
O, %	41.968	39.753	38.038	39.819
Real density, g/cm^3^	1.4139	1.3759	1.3236	1.3632
Volumetric weight, g/cm^3^	0.253	0.375	0.365	0.598

**Table 2 materials-16-01036-t002:** Physical–chemical characteristics of the ACMP obtained in 3.3 kW equipment, with the porosimetric and textural parameters determined after processing the N_2_ sorption isotherms.

	ACMP	Walnut Shells							Peach Kernels	Apricot Kernels	Olive Kernels
ID Characteristic		1	2	3	4	5	6	7	8	9	10
Wa, %		1.43	1.43	1.68	0.51	0.98	1.63	0.9	1.92	2.05	1.87
Ash (db), %		2.17	2.41	2.5	3.03	2.03	2.09	2.79	1.2	2.44	1.34
Volatile (db), %		17.51	9.2	7.02	7.63	11.25	4.48	4.76	6.84	11	5.51
Fixed carbon (db), %		80.32	88.39	90.48	89.34	86.72	93.43	92.45	91.96	86.56	93.15
Real density, g/cm^3^		1.4287	1.5449	1.5562	1.5549	1.4785	1.614	1.5921	1.5504	1.4961	1.5964
Volumetric weight, g/cm^3^		0.382	0.384	0.389	0.356	0.31	0.339	0.333	0.32	0.377	0.368
C, %		79.244	82.534	84.913	84.181	82.456	84.765	83.448	86.587	82.847	79.138
H, %		2.960	1.898	1.772	1.926	2.478	1.411	1.588	1.898	2.457	1.650
N, %		0.577	0.511	0.529	0.552	0.622	1.246	0.623	0.686	1.302	0.577
O, %		2.725	2.654	8.771	6.308	12.977	6.144	2.853	3.288	2.462	7.464
Iodine index, mg/g		274.89	417.11	455	440.8	383.92	469.27	438.44	488.37	303.38	397.34
S_BET_, m^2^/g		33.081	172.762	295.192	252.456	70.776	251.039	138.624	231.749	11.353	114.373
Sµ, m^2^/g		12.749	129.092	160.030	117.725	12.001	127.652	62.256	154.852	0	55.336
Vµ, cm^3^/g		0.005	0.053	0.066	0.044	0.004	0.052	0.024	0.066	0	0.022
Vtot, cm^3^/g		0.023	0.091	0.148	0.133	0.048	0.096	0.078	0.123	0.012	0.074

Acronyms: S_BET_ = surface area BET; Sµ = surface occupied by the micropores volume; Vµ = micropore volume; V_tot_ = total pore volume.

**Table 3 materials-16-01036-t003:** Thermal parameters obtained by thermal pyrolysis of walnut shells, peach, apricot, and olive kernels in an inert gas atmosphere (N_2_).

Waste Type	Heating Rate	Decomposition Stage	T_onset_	T_peak_	W	T_10_	T_20_
	°C/min		°C	°C	%	°C	°C
Walnut shells	10	I	48	88	3.01	253	287
		II	207	-	5.16		
		III	254	339	52.02		
		IV	410	-	12.61		
		Residue			27.2		
Peach kernels	10	I	54	82	2.14	268	296
		II	242	290	23.38		
		III	327	353	32.87		
		IV	404	-	13.37		
		Residue			28.24		
Apricot kernels	10	I	49	78	2.8	258	295
		II	163	-	22.42		
		III	323	344	25.44		
		IV	375	391	26.04		
		Residue			23.3		
Olive kernels	10	I	42	85	2.48	257	284
		II	231	278	23.46		
		III	317	336	28.98		
		IV	357	-	18.67		
		Residue			26.41		

T_onset_ = temperature at which thermal degradation begins in a certain stage; T_peak_ = temperature at which the rate of degradation is maximum; T_10_, T_20_ = temperature corresponding to mass losses of 10%, respectively 20%; W% = mass losses.

**Table 4 materials-16-01036-t004:** Operational parameters run during the laboratory experiments.

Test ID	Waste Type	Maximum Temp., °C ^(1)^	Heating Rate, °C/min	Time, min. ^(2)^	Power Density MW, W/g	
					Initial	Final
T1	Walnut shells	600	14.7	60	3.07	12.56
T2		650	11.91	15	2.94	11.79
T3		630	30	45	5.72	12.57
T4		650	21	45	5.72	11.46
T5		700	29.05	45	5.29	10.05
T6		755	24.06	20	5.97	25.00
T7		650	10.86	22	5.75	23.74
T8	Peach kernels	650	35.12	50	4.80	18.97
T9	Apricot kernels	650	113.8	60	4.70	17.45
T10	Olive kernels	650	41.6	45	4.94	18.75

^(1)^ Temperature at which the process reaches a plateau; ^(2)^ Soaking time at maximum temperature.

**Table 5 materials-16-01036-t005:** Statistical parameters for the porosimetric characteristics determined by SBET analysis.

	Iodine Index, mg/g	SBET, m^2^/g	Sµ, m^2^/g	Vµ, cm^3^/g	Vtot, cm^3^/g
Iodine index, mg/g	1				
S_BET_, m^2^/g	0.845563177	1			
S_µ_, m^2^/g	0.767467391	0.9437926	1		
V_µ_, cm^3^/g	0.752376997	0.9300672	0.9979294	1	
V_tot_, cm^3^/g	0.808159814	0.9606189	0.90532	0.8852413	1

## Data Availability

Not applicable.

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
