# Peer review of "Green Synthesis of Advanced Carbon Materials Used as Precursors for Adsorbents Applied in Wastewater Treatment"

_materials, 2023, doi:10.3390/ma16031036_

Round 1

Reviewer 1 Report

This article introduces the synthesis strategy of advanced carbon materials and their applications. Such ACMPs are not new but represent clearly a sub-domain of ACMPs research that has gained recently an increasing interest in many fields such as water or air purification, catalysis, energy storage and conversion, etc. This review is therefore timely. However, there are still important points, as well as mistakes, that need to be corrected prior to publication.

1.         The review is organized into different parts, from the synthesis approaches to the different potential applications. A general discussion about the main advantages and current limitations of ACMPs in comparison with their homometallic counterparts. Synergestic points should be better discussed, for instance at the end of the paper.

2. Authors shall better discuss the impact of the metal doping on the stability. For instance introducing a lower valence metal cation into a robust ACMPs can lead to a significant decrease of the chemical stability or alternatively introducing a more inert cation (Cr or Ni) or a higher valence metal (e.g. Ti(IV)) can, on the contrary, improve it.

3. Same comment for the challenges associated to a fine characterization of the presence of the ACMPs; a summary of the main recent progresses and remaining challenges of the usual characterization techniques (PDF, EXAFS, TEM, tomography…) should be given including recent publications.

4.  Another application should be at least briefly discussed: magnetism as it is clearly a domain where metal substitution is relevant. Adsorbents are the core agents involved in adsorption process, and they confer process efficiency by means of special technical characteristics, some updated refs could be cited.

5. The author's English is somehow a bit loose which makes the reading quite difficult. This also includes repetitions, grammatical errors, etc. Please improve the quality of the writing accordingly.

Author Response

Reviewer 1

Comments and Suggestions for Authors

1.This article introduces the synthesis strategy of advanced carbon materials and their applications. Such ACMPs are not new but represent clearly a sub-domain of ACMPs research that has gained recently an increasing interest in many fields such as water or air purification, catalysis, energy storage and conversion, etc.

Answer 1: We have proposed to publish an article having 3 main objectives: i) characterization of raw material-waste biomass, ii) carbonization of raw material by microwave low energy technology and iii) characterization of the obtained products ACMP, having in mind their further application in water purification. Considering the aspects related to this specific target application, the technological parameters aimed at obtaining a suitable quality of ACMP, respectively an adequate adsorption capacity.

The other directions developed worldwide, i.e. catalysis, energy storage and conversion, other, are not the subject of our paper. However, the possible use of carbon-based materials in such application (catalysis) are only briefly mentioned, the subject being a matter of another paper.

2.This review is therefore timely. However, there are still important points, as well as mistakes, that need to be corrected prior to publication.

The review is organized into different parts, from the synthesis approaches to the different potential applications. A general discussion about the main advantages and current limitations of ACMPs in comparison with their homometallic counterparts. Synergestic points should be better discussed, for instance at the end of the paper.

Answer 2: the possible use of carbon-based materials in different potential application (catalysis) are only briefly mentioned, but the subject is a matter of another paper.

  1. Authors shall better discuss the impact of the metal doping on the stability. For instance introducing a lower valence metal cation into a robust ACMPscan lead to a significant decrease of the chemical stability or alternatively introducing a more inert cation (Cr or Ni) or a higher valence metal (e.g. Ti(IV)) can, on the contrary, improve it.

Answer 3: The same as answer 2.

  1. Same comment for the challenges associated to a fine characterization of the presence of the ACMPs; a summary of the main recent progresses and remaining challenges of the usual characterization techniques (PDF, EXAFS, TEM, tomography…) should be given including recent publications.

Answer 4: The characterization of ACMP was completed by TEM analysis. This mention was done in manuscript including the abstract.

However, we agree that a more complex characterization would have been interesting, but the authors limited only to the most relevant analysis for the paper topic using the equipment found in the affiliated institutions.To investigate the morphology of the prepared lignocellulosic waste-derived carbonaceous materials at nanoscale, transmission electron microscopy has been considered. Thus, a Hitachi High-TechHT7700 Transmission Electron Microscope (TEM) has been used for registering micrographs for each sample. The equipment has been used in high-contrast mode at 120 kV accelerating voltage. Before analysis, all samples were grinded and dispersed in ethanol by ultrasonication, and later, a few drops were placed on formvar-coated copper grids. The cooper grids thus prepared were then dried overnight at 333 K. Figure 6. Representative TEM (bar scale 100 nm) images (inset: SAED patterns) of the waste type analyzed: a) walnut shells; b) peach kernels; c) apricot kernels; d) olive kernels. Thus, Figure 6 shows TEM images revealing the morphology of the prepared carbona-ceous materials from walnut shells, peach, apricot, and olive kernels. As could be seen, all TEM images indicate the development of a porous structure of mainly amorphous carbon phase, confirmed by SAED patterns (insets in Figure 6), with regions display-ing structure ordering, most probably due to the formation of graphene layers. Moreover, the SAED patterns also reveal the presence of graphite crystals or another form of carbon with a lower degree of order.Additional reference was enclosed in the manuscript:

Cheng, J.; Xu, Q.; Wang, X.; Li, Z.; Wu, F.; Shao, J.; Xie, H. Ultrahigh-surface-area nitrogen-doped hierarchically porous carbon materials derived from chitosan and betaine hydrochloride sustainable precursors for high-performance supercapacitors. Sustain. Energy Fuels, 2019, 3, 1215–1224.

  1. Another application should be at least briefly discussed: magnetism as it is clearly a domain where metal substitution is relevant.

Answer 5: preparation and application of magnetic biochar in water treatment was not the subject of our paper. However, we agree that the subject is challenging, and many publications are available.

  1. Adsorbents are the core agents involved in adsorption process, and they confer process efficiency by means of special technical characteristics, some updated refs could be cited.

Answer 6: Some relevant references (new listed as 4-17) have been mentioned and inserted in the manuscript, please find below.

  1. Cha, J.S.;Park, S.H.;Jung, S.-C.; Ryu, C.;Jeon, J.-K.;Shin, M.-C.;Park,Y.-K. Production and utilization of biochar: A review. Journal of Industrial and Engineering Chemistry, 2016, 40, 1-15.
  2. Tang, W.; Zhang, Y.; Zhong, Y.; Shen T.; Wang, X.; Xia, X.; Tu, J. Natural biomass-derived carbons for electrochemical energy storage, Materials Research Bulletin, 2017, 88, 234-241.
  3. Sweetman, J.;  May, S.;  Mebberson, N.;  Pendleton, P.; Vasilev, K.; Plush, S.E.; Hayball, J.D. Activated carbon, carbon nanotubes and graphene: materials and composites for advanced water purification, Journal of Carbon Research, 2017, 3, 1-29.
  4. Frank, J.; Ruhl, A.S.; Jekel, M. Impacts of backwashing on granular activated carbon filters for advanced wastewater treatment. Water Res. 201587, 166–174.
  5. Gibert, O.; Lefèvre, B.; Fernández, M.; Bernat, X.; Paraira, M.; Pons, M. Fractionation and removal of dissolved organic carbon in a full-scale granular activated carbon filter used for drinking water production. Water Res. 201347, 2821–2829.
  6. Moreno-Castilla, C. Adsorption of organic molecules from aqueous solutions on carbon materials. Carbon 200442, 83–94.
  7. Smith, S.C.; Rodrigues, D.F. Carbon-based nanomaterials for removal of chemical and biological contaminants from water: A review of mechanisms and applications. Carbon 201591, 122–143.
  8. Bhatnagar, A.; Hogland, W.; Marques, M.; Sillanpää, M. An overview of the modification methods of activated carbon for its water treatment applications.  Eng. J. 2013219, 499–511.
  9. Richardson, S.D.; Kimura, S.Y. Water analysis: Emerging contaminants and current issues.  Chem. 201688, 546–582.
  10. Pavoni, B.; Drusian, D.; Giacometti, A.; Zanette, M. Assessment of organic chlorinated compound removal from aqueous matrices by adsorption on activated carbon. Water Res. 200640, 3571–3579.
  11. Predeanu, G.; Axinte, S.M.; Drăgoescu, M.F.; González, Z.; Álvarez, P.; Granda, M.; Menéndez, R.; Fiti, A.; Acevedo, B.; Melendi-Espina, S.; Gryglewicz, G.; Fernández, J.J.; Slăvescu, V. Microwave heating as a novel route for obtaining carbon precursors from anthracene oil. Fuel Processing Technology, 2019, 192, 250–257.    
  12. Gonzáles, Z.; Acevedo, B.; Predeanu, G.; Axinte, S.M.; Drăgoescu, M.F.; Slăvescu, ;  Fernandez, J.J.; Granda, M.; Gryglewicz, G.; Melendi-Espina, S. Graphene materials from microwave-derived carbon precursors. Fuel Processing Technology, 2021, 217, 1-7.
  13. Malode, S.J.; Shanbhag, M.M.; Kumari, R.; Dkhar, D.S.; Chandra, P.; Shetti, P.N. Biomass-derived carbon nanomaterials for sensor applications. Journal of pharmaceutical and biomedical analysis, 2023, 222, 1-24.
  14. Liu, P.; Wang, Y.; Liu, J. Biomass-derived porous carbon materials for advanced lithium sulfur batteries, Energy Chem. 2019, 34, 171–185.
  15. The author's English is somehow a bit loose which makes the reading quite difficult. This also includes repetitions, grammatical errors, etc. Please improve the quality of the writing accordingly.

Answer 7: Some of the paragraphs have been rewritten and the English was improved.

Reviewer 2 Report

Specific comments:

3. Results and discussion.

3.3.3. Scanning electron microscopy (SEM) analysis

Page 10, at the end of paragraph 3.3.3, the authors argue, “The sample does not show a porous structure, and this fact can be argued by pores clogging after the grinding process “, this statement is not sufficiently supported, so to reinforce and clarify their interpretation of the possible origin of the morphology of the sample, complement information on sample preparation for SEM analysis and compare with other samples.

Page 11, fig 5. It is recommended that the SEM images include supplemental information, such as, magnification, window size, etc.,

Page 11, the author claim, “The photomicrographs show the presence of typical porous morphology having pores of approximately 0.25 μm”, it is possible to see size and shape of particles by simple inspecting the images in figure 5, and it is not possible to state that the pores are approximately 5 mm, carefully review this statement and justify the answer.

Page 12, Clarify the definition of “yields”.

Page 13, In the last paragraph the value reported for sample 10 is lower than that of sample 6, (93.43 %), see table 2.

Page 14, in section 3.4.1. Adsorption characteristics, it is suggested to include a comparison with the values reported in the literature by other authors and works.

Page 4, in the section Materials and methods, subsection, 2.5. BET analysis and adsorption processes, the authors describe the BET analysis technique; however, they do not present results of this technique, for example, isotherms and BET analysis of the samples are not presented.

Careful, in the document there are two subsections with the same number, (3.4.1) review and correct.

In section 3 Conclusions, it is suggested to review and improve the last paragraph on page 17.

It is recommended to review and correct for acceptance

Author Response

Reviewer 2

Comments and Suggestions for Authors

  1. Results and discussion.

3.3.3. Scanning electron microscopy (SEM) analysis

  1. Page 10, at the end of paragraph 3.3.3, the authors argue, “The sample does not show a porous structure, and this fact can be argued by pores clogging after the grinding process “, this statement is not sufficiently supported, so to reinforce and clarify their interpretation of the possible origin of the morphology of the sample, complement information on sample preparation for SEM analysis and compare with other samples.

Answer 1: The correction was done according to the reviewer’s suggestion: “The sample does not show a visible native porous structure, and this fact can be argued by pores clogging after the grinding process“.

The existence of a native porosity in the raw material depends of course on the type of raw material, and the parameter “real density” highlights it as well (Table 1). During heating the evolution of the volatiles create pores in the compact walls of the material, which are also initiated by the original porosity of the native material.

SEM analysis information regarding sample preparation were enclosed in manuscript, Chap. 2. Materials and methods, sub-chap. 2.4, as follows: 

The preparation of the raw material for SEM analysis followed several steps, as follows: 1. Washing: samples were washed 3 times with a sufficient volume of distilled water (1L water/100g waste, under agitation ( for 15 minutes) at a temperature of 40 °C; 2. Drying: samples were dried in a heating oven at 105 °C for 24 hours after which being transferred to a desiccator; 3. Primary crushing; 4. Grinding: using the Pulverisette 11 mill in 3 stages i) 5 minutes at 3500 rpm for 12 s and a 20 s break; ii) 5 minutes at 5500 rpm for 15 s and a 60 s break; iii) 5 minutes at 9000 rpm for 20 s and a 60 s break; 5. Sizing: for 15 minutes using the Retsch AB 200 system (amplitude 55) on the following sieves: 4mm; 2mm; 1mm; 0.5mm; 0.25 mm; 0.125mm. The fraction between 125 µm and 250 µm was used for the analyses.

  1. Page 11, fig 5. It is recommended that the SEM images include supplemental information, such as, magnification, window size, etc.,

Answer 2: SEM images including supplementary information during the analysis, such as magnification, size, etc., were enclosed in the manuscript, please, find new Fig. 5.

A

D

C

B

  1. Page 11, the author claim, “The photomicrographs show the presence of typical porous morphology having pores of approximately 0.25 μm”, it is possible to see size and shape of particles by simple inspecting the images in figure 5, and it is not possible to state that the pores are approximately 5 mm, carefully review this statement and justify the answer.

Answer 3: By enlarging the image of Fig. 5, one could detect very fine pores on the surface of the raw biomass particles (Fig. 5, C), which are estimated at approximately 1 µm.

The error has been corrected in the manuscript, such as: “The photomicrographs show the presence of typical porous morphology of biomass waste having pores of about 1 μm”. 

  1. Page 12, Clarify the definition of “yields”.

Answer 4: The yield of a reaction is the ratio of the desired product formed (in %) to the total starting amount. In our case yields refers to the solid, liquid, and gaseous amounts (in %) related to the total starting waste amount before carbonization. From a total 100 g of lignocellulosic waste, by conventional pyrolysis global yields such as: solid biochar 30%, tar 42% and gases of 28% are obtained.

  1. Page 13, In the last paragraph the value reported for sample 10 is lower than that of sample 6, (93.43 %), see table 2.

Answer 5: The error was corrected.

  1. Page 14, in section 3.4.1. Adsorption characteristics, it is suggested to include a comparison with the values reported in the literature by other authors and works.

Answer 6: In sections 3.4.1 and 3.4.2 supplementary information are enclosed in the manuscript, for the adsorption characteristics (iodine index and surface area BET) that were compared with the values reported in the literature (new listed as 85-90) by other authors and works, please find them below.

  1. Lee, Y.; Eum P-R-B.; Ryu, C.; Park, Y.-K.; Jung J.-H.; Hyun S. Characteristics of biochar produced from slow pyrolysis of Geodae-Uksae. Bioresource technology, 2013, 130, 345-350.
  2. Sun, J.; He, F.; Pan, Y.; Zhang, Z. Effects of pyrolysis temperature and residence time on physicochemical properties of different biochar types. Acta Agriculturae Scandinavica, Section b—Soil & Plant science, 2017, 67, 12–22.
  3. Lawal, A.; A.; Hassan, M.; A.; Zakaria, M.; R.; Yusoff, M.; Z.; M.; Norrrahim, M.; N.; F.; Mokhtar, M.; N.; Shirai, Effect of oil palm biomass cellulosic content on nanopore structure and adsorption capacity of biochar. Bioresource Technology, 2021, 332, 125070.
  4. Jian, X.; Zhuang, X.; Li, B.; Xu, X.; Wei, Z.; Song, Y.; Jiang Comparison of characterization and adsorption of biochars produced from hydrothermal carbonization and pyrolysis. Environmental Technology & Innovation, 2018, 10, 27-35.
  5. Park, J.; H.; Ok, Y.; S.; Kim, S.; H.; Kang, S.; W.; Cho, J.; S.; Heo, J.; S.; Delaune, R.; D.; Seo, D.; C. Characteristics of biochars derived from fruit tree pruning wastes and their effects on lead adsorption. J Korean Soc Appl Biol Chem. 2015, 1-10.
  6. Page 4, in the section Materials and methods, subsection, 2.5. BET analysis and adsorption processes, the authors describe the BET analysis technique; however, they do not present results of this technique, for example, isotherms and BET analysis of the samples are not presented.

Answer 7: BET analysis results are enclosed in Table 2 for SBET=Surface area BET; Sµ= surface occupied by the micropores volume; Vµ=micropores volume; Vtot= total pores volume.

According to the reviewer comments, additional information and references were enclosed (new listed as 91) in the manuscript, conclusions and abstract, as follows:

“Figure 6 presents the Langmuir isotherms (A1, B1) and the corresponding pore size distribution (A2, B2) of ACMP made from walnut shells (A) and peach kernels (B) obtained when the adsorption is restricted to the monolayer level. According to IUPAC classification [91], the registered isotherms are of type I (b) which usually are found in carbonaceous materials having pore size distributions in micro-meso range (wider micropores and narrow mesopores). Thus, the ACMP adsorbent seems to have micropores that are too small to support the adsorption of nitrogen molecules in more than one layer, as well as the capillary condensation in the pores. Both ACMP samples (walnut shells and peach kernels) exhibit the same comparable type I (b) isotherm.

A1

B2

B1

A2

Figure 6. Nitrogen adsorption-desorption isotherms (A1, B1) and the corresponding pore size distribution (A2, B2) obtained for ACMP walnut sample (A) and peach kernels (B).

  1. Thommes, M.; Kaneko, K.; Neimark, A.V., Olivier, J.P.; Rodriguez-Reinoso, F.; Rouquerol, J.; Sing, K.S.W. Physisorption of gases, with special reference to the evaluation of surface area and pore size distribution (IUPAC Technical Report). Pure Appl Chem. 2015, 87 (9-10), 1051-1069.

The pore size distributions in micro-meso range (wider micropores and narrow mesopores) developed in ACMP by microwave heating give the possibility of assessing these materials for the intended applications. The results are promising regarding the ACMP addressed to purification of waters infested for example with certain dyes having molecule sizes in the range of ACMPs pore dimensions below 2 nm, such as: methylene blue, rhodamine, methyl orange or others. Also, tests are foreseen regarding removal of ions with pore dimensions below 1 nm such as nitrate and sulphate ions widely used industrially and found in underground waters (NO3) or waste waters (SO2−4).

  1. Careful, in the document there are two subsections with the same number, (3.4.1) review and correct.

Answer 8: The error has been corrected.

  1. In section 3 Conclusions, it is suggested to review and improve the last paragraph on page 17.

Answer 9: The paragraph was reviewed and improved, such as:

The morphological aspects of ACMP identified by optical microscopy can be correlated with the iodine adsorption capacity developed during degassing at 600-700° C, as follows: it has a specific microstructure, characterized by a small porosity created in the wider pore walls, with a cellular structure consisting of rounded pores. The increase of intragranular porosity occurs in parallel with the reduction of the carbon matrix and of the volume of pore walls.The photomicrographs highlight both the characteristics of the porous texture and the types of carbon matrix, which can contribute, in certain proportions, to the dimensions of the adsorption surface and in the subsequent phases of ACMP activation.

It is recommended to review and correct for acceptance

Submission Date

20 December 2022

Round 2

Reviewer 2 Report

No comments